# Glutamine Availability Controls BCR/Abl Protein Expression and Functional Phenotype of Chronic Myeloid Leukemia Cells Endowed with Stem/Progenitor Cell Potential

**DOI:** 10.3390/cancers13174372

**Published:** 2021-08-30

**Authors:** Martina Poteti, Giulio Menegazzi, Silvia Peppicelli, Ignazia Tusa, Giulia Cheloni, Angela Silvano, Caterina Mancini, Alessio Biagioni, Alessandro Tubita, Nathalie M. Mazure, Matteo Lulli, Elisabetta Rovida, Persio Dello Sbarba

**Affiliations:** 1Department of Experimental and Clinical Biomedical Sciences, University of Florence, Viale G.B. Morgagni 50, 50134 Firenze, Italy; martina.poteti@gmail.com (M.P.); g.menegazzi@student.unisi.it (G.M.); silvia.peppicelli@unifi.it (S.P.); Ignazia.tusa@unifi.it (I.T.); gcheloni@bidmc.harvard.edu (G.C.); angela.silvano@hotmail.it (A.S.); caterina.mancini1995@gmail.com (C.M.); alessio.biagioni@unifi.it (A.B.); alessandro.tubita@unifi.it (A.T.); matteo.lulli@unifi.it (M.L.); 2Beth Israel Deaconess Medical Center, Department of Medicine, Division of Genetics, Harvard University Medical School, 330 Brookline Avenue, Boston, MA 02215, USA; 3Mediterranean Centre for Molecular Medicine-INSERM U1065, University of Nice-Sophia-Antipolis, 151 Route Saint Antoine de Ginestière, 06204 Nice, France; Nathalie.Mazure@unice.fr

**Keywords:** chronic myeloid leukemia, low oxygen, glutamine, glucose, stem cell niche, minimal residual disease

## Abstract

**Simple Summary:**

In chronic myeloid leukemia (CML), a neoplasm brilliantly taken care of by a molecularly targeted therapeutic approach, the achievement of cure is nevertheless prevented by the maintenance of a small subset of treatment-resistant leukemia stem cells (LSCs), sustaining the so-called minimal residual disease of CML. The phenotypical and functional characterization of this LSC subset is, therefore, crucial to aim at the eradication of disease. Such a characterization includes the acquisition of information relative to the metabolic profile of treatment-resistant LSCs, which is functional to their maintenance in bone marrow. A number of metabolic features of LSCs were shown to determine their sensitivity or resistance to therapy. Glutamine metabolism emerged from this study as a potential target to overcome the persistence of therapy-resistant LSCs.

**Abstract:**

This study was directed to characterize the role of glutamine in the modulation of the response of chronic myeloid leukemia (CML) cells to low oxygen, a main condition of hematopoietic stem cell niches of bone marrow. Cells were incubated in atmosphere at 0.2% oxygen in the absence or the presence of glutamine. The absence of glutamine markedly delayed glucose consumption, which had previously been shown to drive the suppression of BCR/Abl oncoprotein (but not of the fusion oncogene *BCR*/*abl*) in low oxygen. Glutamine availability thus emerged as a key regulator of the balance between the pools of BCR/Abl protein-expressing and -negative CML cells endowed with stem/progenitor cell potential and capable to stand extremely low oxygen. These findings were confirmed by the effects of the inhibitors of glucose or glutamine metabolism. The BCR/Abl-negative cell phenotype is the best candidate to sustain the treatment-resistant minimal residual disease (MRD) of CML because these cells are devoid of the molecular target of the BCR/Abl-active tyrosine kinase inhibitors (TKi) used for CML therapy. Therefore, the treatments capable of interfering with glutamine action may result in the reduction in the BCR/Abl-negative cell subset sustaining MRD and in the concomitant rescue of the TKi sensitivity of CML stem cell potential. The data obtained with glutaminase inhibitors seem to confirm this perspective.

## 1. Introduction

Chronic myeloid leukemia (CML) is a myeloproliferative disorder driven by the reciprocal translocation t(9;22)(q34;q11) in hematopoietic stem cells (HSCs) [1]. The translocation creates the fusion oncogene *BCR*/*abl*, the product of which is the BCR/Abl constitutively active cytoplasmic tyrosine kinase. The introduction, almost two decades ago, of tyrosine kinase inhibitors (TKi) capable of blocking BCR/Abl kinase activity strikingly improved the management of CML [2]. However, despite a brilliant response to TKi treatment as for the induction of remission, many patients undergo a relapse of disease upon treatment withdrawal or even under continued therapy. Relapse is apparently due to the persistence, in CML patients, of TKi-insensitive leukemia stem cells (LSCs) sustaining therapy-resistant minimal residual disease (MRD) [3].

LSCs, similarly to HSCs, are believed to be selectively hosted in stem cell niches (SCNs) of bone marrow (BM), sites that are physiologically characterized by, among other features, a very low oxygen tension. In SCNs, stem cell potential is preserved and HSCs/LSCs are protected from stimuli boosting their clonal expansion [4,5,6,7,8], yet are allowed to cycle [9,10]. Studies carried out in our laboratory showed that the incubation of CML cells in low oxygen time-dependently suppresses the BCR/Abl oncoprotein (but not the *BCR*/*abl* oncogene) and that a cell subset endowed with stem cell potential is capable of persisting in low oxygen independently of BCR/Abl signaling [11,12]. Among a number of different possible pathways to the resistance to TKi of cells sustaining the risk of relapse of disease, the most straightforward is arguably the lack of the molecular target of TKi in *BCR*/*abl*-positive/BCR/Abl-negative LSCs where BCR/Abl expression can be later rescued, then boosting clonal expansion. An initial characterization of the metabolic mechanisms driving BCR/Abl suppression showed that it occurs when glucose approaches complete exhaustion, which is obviously made easier when cells reside in a low oxygen environment [12]. We proposed, on this basis, a model of low-oxygen SCN where BCR/Abl-positive or -negative LSC subsets are spatially distributed according to local substrate availability and in function of their different metabolic profiles [13]. In SCN zones where glucose is available, BCR/Abl expression would predispose LSCs to clonal expansion, whereas zones under glucose shortage would host LSCs adapted to persist independently of BCR/Abl signaling and thereby, treatment-resistant MRD. In this scenario, the deepening of conditions controlling the balance of BCR/Abl expression-related phenotypes within the SCN is of obvious interest.

In this study, we addressed the role of the availability of glutamine in the control of BCR/Abl expression in CML cells incubated in atmosphere at very low oxygen tension. Besides glucose, glutamine is the major substrate supporting energy production and biosynthesis in mammalian cells [14], and both glucose and glutamine control cell survival and cycle progression through independent, although connected, bioenergetic pathways [15,16]. Glutamine also plays a key role in low oxygen, and glutamine starvation has been shown to interfere with glucose catabolism and to weaken cell adaptation to low oxygen [17,18].

To characterize the regulative role of glutamine availability on CML cells, we took advantage of the precise definition of culture conditions where glucose is progressively and rapidly consumed in a function of incubation time in low oxygen. This experimental system is taken as a reasonable surrogate to the glucose gradient occurring in vivo in a function of distance from blood supply [11,12], which governs the spatial compartmentalization of LSC subsets in relation to their metabolic profile [13]. In such a system, the metabolic context was varied simply by adding, or not, glutamine to glucose-supplemented culture medium. The physiological relevance of glutamine deprivation in vitro can be traced back to the strong depletion of glutamine observed within tumor cell masses in vivo [19]. The results obtained demonstrated that glutamine availability, via the control of glucose consumption and utilization, is key to the regulation of BCR/Abl expression/signaling in low oxygen, i.e., to the balance between the pools of TKi-sensitive or -resistant cells endowed with stem/progenitor cell potential. The translational perspective emerging from this study is underscored.

## 2. Materials and Methods

### 2.1. Cells and Culture Conditions

The K562 [20] and KCL22 [21] BCR/Abl-positive cell lines derived from CML patients in blast crisis were routinely cultured in RPMI 1640 medium supplemented with 10% fetal bovine serum, 2 mM glutamine, 50 units/mL penicillin and 50 mg/mL streptomycin (all from EuroClone; Paignton, UK) and incubated at 37 °C in a water-saturated atmosphere containing 21% O_2_ and 5% CO_2_. Experiments were performed with cells from exponentially growing maintenance cultures replated at 3 × 10^5^/mL and incubated at 37 °C in a water-saturated, low-oxygen atmosphere containing 0.2% O_2_, 5% CO_2_ and 95% N_2_, in a gas-tight manipulator/incubator (DG250 Anaerobic Workstation; Don Whitley Scientific; Shipley, Bridgend, UK). Cell viability was measured using a trypan blue (#F-7378, Sigma-Aldrich; St. Louis, MO, USA; 0.2 g in 99.8 mL water) exclusion test. Glucose and lactate concentrations in culture medium were measured using a YSI 2300 STAT Plus analyzer (YSI Life Sciences; Yellow Springs, OH, USA) and pH changes in culture medium with a PHM61 pH meter (Radiometer; Copenhagen, Denmark).

### 2.2. Reagents

2-Deoxy-D-glucose (2-DG), a competitive inhibitor of hexokinase and thereby glycolysis [22] (#154176, Calbiochem; San Diego, CA, USA) and metformin, an inhibitor of mitochondrial complex I [23] (#317240, Calbiochem) were dissolved in water and added to cultures at 1 mM. 3-(3-pyridinyl)-1-(4-pyridinyl)-2-propen-1-one (3PO), a selective inhibitor of 6-phosphofructo-2-kinase/fructose-2,6-bisphosphatase (PFKFB3) and thereby of glucose uptake and glycolytic flux [24], was dissolved in dimethylsulfoxide (DMSO) and added to cultures at 10 mM (HY-19824, MedChemExpress; Sollentuna, Sweden). 6-amino-nicotinamide (6-AN), an inhibitor of 6-phosphogluconate dehydrogenase and thereby of pentose phosphate pathway (PPP) [25], was dissolved in DMSO and added to cultures at 29 mM (#329895, Cayman Chemical Company; Ann Arbor, MI, USA.). Bis-2-(5-phenyl-acetamido-1,2,4-thiadiazoyl-2-yl)ethyl-sulfide (BPTES), an inhibitor of kidney-type glutaminase GLS1 [26] was dissolved in DMSO and added to cultures at 10 mM (HY-12683, MedChemExpress). Telaglenastat (TGS), another GLS1 inhibitor, was dissolved in DMSO and added to cultures at 0.5 mM (HY-12248, MedChemExpress), the BCR/Abl-active TKi Imatinib-mesylate (IM), a tyrosine kinases inhibitor, was dissolved in H_2_O and added to cultures at 1 mM (HY-50946 MedChemExpress).

### 2.3. Cell Lysis and Western Blotting

Cells were lysed in Laemmli buffer, lysates clarified by centrifugation (20,000 g, 10 min, room T) and protein concentration in supernatants determined using the BCA method. Equal protein aliquots (50 mg) were subjected to SDS-PAGE in 9% polyacrylamide gel and then electroblotted onto nitrocellulose membranes (GE Healthcare Life Sciences; Pittsburgh, PA, USA). Membranes were incubated for 1 h at RT in PBS/0.1% BSA (Sigma-Aldrich) and then overnight at 4 °C with the primary antibody in PBS/0.1% BSA/0.1% Tween-20. Primary antibodies (Ab) used were rabbit polyclonal anti-c-Abl (sc-131, Santa Cruz Biotechnology; St. Cruz, CA, USA), diluted 1:750 in PBS/0.1% Tween-20; rabbit polyclonal anti-phospho-Crkl (#3181, Cell Signaling Technology; Danvers, MA, USA), rabbit anti-ARD1 (produced in Dr. Nathalie Mazure’s laboratory) and mouse monoclonal anti-vinculin (V9131, Sigma-Aldrich), all diluted 1:1000 in PBS/0.1% Tween-20. After washing three times with PBS/0.1% Tween-20, membranes were incubated for 1 h at RT in PBS/0.1% BSA containing an IRDye800CW- or IRDye680-conjugated secondary Ab (LI-COR^®^; Lincoln, NE, USA). Mouse and rabbit IRDye800CW-Ab were diluted 1:20,000, goat IRDye680-Ab 1:40,000, mouse and rabbit IRDye680-Ab 1:30,000, goat IRDye680-Ab 1:15000. Ab-coated protein bands were visualized using Odyssey Infrared Imaging System Densitometry (LI-COR^®^).

### 2.4. RNA Isolation and Quantitative Real Time PCR 

Total RNA was extracted from cells by using TRI Reagent (Sigma-Aldrich). The amount and purity of RNA were determined spectrophotometrically. cDNA synthesis was obtained by incubating 1 μg of total RNA with iScript reverse transcription supermix (BioRad, Milan, Italy) according to the manufacturer’s instructions. Quantitative real time PCR (qPCR) was performed using the SsoAdvanced Universal SYBR Green Supermix (BioRad). The qPCR analysis was carried out in triplicate with a CFX96 Real-Time PCR System (BioRad) with the default PCR setting: 40 cycles of 95 °C for 10 s and 60 °C for 30 s. The fold change was determined by the comparative Ct method using 18S and b-actin as reference genes. Primer sequences are listed in Table 1.

### 2.5. Culture Repopulation Ability (CRA) Assay

This non-clonogenic assay allows us the estimate in vitro of the content of a hematopoietic cell population with stem/progenitor cells endowed with Marrow Repopulation Ability in vivo [9]. The CRA assay measures the power of normal or leukemic [10,11,12,27,28,29,30] hematopoietic cells subjected to an experimental treatment (such as, for example, drug treatment or incubation in atmosphere at low oxygen tension) to repopulate unselective cultures enabling maximal growth (incubation at 21% O_2_, absence of drug) where the stem/progenitor cell potential of input cells is exploited. In this study, cells (3 × 10^5^ cells/mL) were incubated for 4 or 7 days in low-oxygen experimental cultures established in the absence or the presence of glutamine and drug-treated or not (liquid culture 1; LC1). Cells were then washed free of drug and transferred (3 × 10^4^ cells/mL) to growth-permissive assay cultures (LC2). The time necessary to reach the peak of LC2 repopulation (and in particular the presence or not of a lag phase before repopulation starts) is a characteristic of the stem/progenitor cell subset transferred from LC1, while the area underlying the LC2 repopulation curve indicates the level of maintenance of stem/progenitor cell potential in LC1 [9,27].

### 2.6. Statistical Analysis

All measures were performed in triplicate. Differences with *p* values ≤ 0.05, as determined using the Student’s *t*-test for paired samples (two-tailed), were considered statistically significant.

## 3. Results

### 3.1. Cell Proliferation and Glucose Consumption Were Reduced in Glutamine-Free CML Cell Cultures

The effects of the absence or the presence of glutamine on the viable cell number in culture were compared first under standard incubation conditions (atmosphere at 21% oxygen). In the absence of glutamine, the cell growth of either K562 or KCL22 cells was reduced (Figure 1A,B), while the decrease in the glucose concentration in culture medium was delayed (Figure 1C,D), indicating that glutamine availability regulates cell proliferation and glucose consumption in our experimental system.

### 3.2. Glucose Consumption and BCR/Abl Protein Suppression in Low Oxygen Were Delayed in the Absence of Glutamine

We previously demonstrated that the BCR/Abl expression is inversely related to the glucose concentration in culture medium [12] when CML cells are incubated in atmosphere with very low oxygen (0.2% O_2_), a condition reproducing a relevant feature that SCNs exhibit in vivo [5,6,7,8]. In low oxygen and in the absence of glutamine, glucose consumption from culture medium was delayed with respect to glutamine-supplemented cultures (Figure 2A,B). In the presence of glutamine, glucose was indeed exhausted relatively rapidly. In keeping with the faster glucose catabolism, glutamine accelerated (significant differences at early times of incubation) the lactate release into (Figure 2C,D) and the acidification of (Figure 2E,F) the culture medium. Furthermore, in the presence of glutamine, BCR/Abl expression and signaling (CRKL phosphorylation) were time-dependently suppressed in low oxygen, more rapidly in K562 than KCL22 cells (Figure 2G,H), as previously reported [4,8]. In the absence of glutamine, the fading-off of BCR/Abl expression/signaling was delayed (Figure 2G, days 3–5; Figure 2H, days 5–7). Overall, the results of Figure 2 indicated that glutamine availability controls glucose consumption and BCR/Abl protein expression and signaling in our experimental system.

### 3.3. Glutamine Controls the Maintenance of Stem/Progenitor Cell Potential in Low Oxygen

We previously showed that the maintenance of stem/progenitor cell potential in low oxygen follows different patterns, related to the maintenance or loss of BCR/Abl expression/signaling [12,27]. Leukemia cell lines, including those used for our study, are well-known to be phenotypically heterogeneous, including immature “blast” cells as well as differentiated cells, which can be even induced to mature under drug pressure [31]. We showed that such a heterogeneity also comprises phenotypes endowed with stem/progenitor cell potential, as well as cell subsets capable of standing an oxygen shortage [10,11,12,29,30]. In the experiments reported here, the cells were incubated in low oxygen in the absence or the presence of glutamine for 7 days, the standard, previously established, incubation time necessary to complete BCR/Abl suppression in control glutamine-supplemented cultures.

The cells rescued from these day seven experimental LC1s were then transferred into assay LC2, incubated at 21% oxygen, and containing glutamine in any case, to determine the maintenance of stem/progenitor cell potential at the end of LC1 by measuring LC2 repopulation (CRA assay). The cells of both lines rescued from control glutamine-supplemented LC1 repopulated LC2 after a 1–2 week-long lag phase (Figure 3A,B), a kinetics typically driven by LC1 cells where BCR/Abl is suppressed [12,27]. This lag phase was absent in the repopulation of LC2 established with KCL22 cells recovered from glutamine-free LC1 (Figure 3B), in keeping with the maintained expression of BCR/Abl at the time of transfer. K562 cells, on the contrary, exhibited identical LC2 repopulation kinetics when rescued from glutamine-supplemented or glutamine-free LC1, most likely due to BCR/Abl suppression in the day seven LC1 in both experimental conditions (Figure 2G). Therefore, in a new set of experiments, K562 cell LC1 were incubated for 4 days, a time point where differences as for BCR/Abl expression and signaling between glutamine-supplemented and glutamine-free cultures were maximal (Figure 2G). The CRA assays of K562 cells from the day four LC1 yielded LC2 kinetics exhibiting (glutamine-supplemented LC1), or not (glutamine-free LC1), a lag phase (Figure 3C), matching the results obtained for the day seven KCL22 cell LC1 (Figure 3B). This indicates that glutamine availability controls not only BCR/Abl expression/signaling, but also the functional phenotype of low oxygen-resistant CML cells endowed with stem/progenitor cell potential.

### 3.4. Inhibition of Glucose Catabolism in the Presence of Glutamine Mimics the Effects of Its Absence

We then determined whether glutamine availability controls BCR/Abl expression/signaling and the maintenance of stem/progenitor cell potential in low oxygen by regulating glucose catabolism. When 2-DG was added to LC1 to inhibit glycolysis, the day four K562 or day seven KCL22 cells repopulated LC2 rapidly, even when rescued from glutamine-containing LC1 (Figure 4A,B). Indeed, 2-DG prevented the suppression of BCR/Abl expression/signaling in the presence of glutamine (Figure 4C,D; lane three vs. seven), mimicking what occurs in the absence of glutamine (lane six). In contrast, the inhibition of cell respiration by metformin did not mimic the effects of the lack of glutamine (Figure 4C,D, lane four vs. five; E,F); actually, metformin treatment of glutamine-supplemented LC1 resulted, with respect to untreated LC1, in a minor (Figure 4E) or more marked (Figure 4F) further delay of LC2 repopulation. These experiments indicated that glycolysis mediates the effects of glutamine on the maintenance of BCR/Abl expression/signaling and stem/progenitor cell potential in low oxygen. This conclusion was confirmed using 3PO, another glycolysis inhibitor. LC1 treatment with 3PO, but not with its solvent DMSO, prevented the suppression of BCR/Abl signaling in the presence of glutamine (Figure 5A,B). 3PO allowed a rapid LC2 repopulation, even in the presence of glutamine (Figure 5C,D). Interestingly, similar results were obtained with 6-AN, an inhibitor of PPP, an alternative way to metabolize glucose (Figure 5A,B,E,F). Thus, the effects of glutamine on BCR/Abl signaling and the maintenance of stem/progenitor cell potential in low oxygen are mediated by glucose consumption irrespective of the metabolic pathway that drives glucose catabolism.

### 3.5. Glutamine-Dependent Glucose Exhaustion Is Counterbalanced by Respiration Induction

The findings that, in the presence of glutamine, glucose was exhausted at the end of incubation in low oxygen (Figure 2A,B) and that metformin treatment affected LC2 repopulation appreciably (Figure 4E,F) prompted us to test the effects of glutamine on the expression of the markers of mitochondrial respiration and biogenesis. Glutamine was found necessary for, or to markedly enhance, the expression of cytochrome-C and cytochrome-oxidase-5B, markers of respiratory activity, as well as of PGC-1α, a marker of mitochondrial biogenesis (Figure 6). Thus, the glutamine-dependent maintenance in the low-oxygen LC1 of the delayed LC2-repopulating stem/progenitor cell potential seems to be sustained by a cell subset characterized by a respiration-oriented metabolic profile.

### 3.6. Inhibition of Glutamine Metabolism by BPTES Mimics the Effects of the Absence of Glutamine 

To determine whether the effects of glutamine shortage observed in vitro are liable to be reproduced in vivo, where glutamine is obviously always present, we tested the effects of BPTES, an inhibitor of glutamine conversion to glutamate and thus, likely, of the biological activity of glutamine. K562 or KCL22 cells were incubated in low oxygen for 4 or 7 days, respectively, in the absence or the presence of glutamine and/or BPTES. The addition of BPTES, but not of its solvent DMSO, to glutamine-supplemented cultures resulted in the maintenance of BCR/Abl signaling (Figure 7A,B; lane five vs. two) as well as of the stem/progenitor cell potential responsible for rapid LC2-repopulation (Figure 7C,D). These results indicated that BPTES treatment mimics the effects of glutamine shortage, and also that in our system, the regulatory role of glutamine is mediated via its conversion to glutamate. 

### 3.7. Glutaminase Inhibitors Sensitize CML Cells to TKi Treatment

As summarized in the Introduction, BCR/Abl suppression makes CML cells endowed with stem/progenitor cell potential refractory to the action of TKi because they are devoid of the TKi molecular target. On this basis, we explored the translational value of the results obtained with BPTES by testing whether glutaminase inhibitors, forcing stem/progenitor cell potential to remain BCR/Abl-dependent, may increase its sensitivity to TKi. Figure 8 shows that the delayed LC2-repopulating stem/progenitor cell potential of LC1 treated with IM (the prototype of TKi) was not or transiently reduced, as expected, with respect to that of untreated cells. On the contrary, when BPTES or TGS were added to LC1 together with IM, a marked reduction in the LC2 repopulation was obtained.

## 4. Discussion

This study showed that glutamine is a critical regulator of BCR/Abl signaling in low oxygen, via the control of glucose consumption from culture medium, which we had previously shown to drive BCR/Abl protein suppression [12]. The relationships of glucose consumption to, on one hand, *BCR/Abl*-induced transformation [32] and, on the other, glutamine metabolism in cancer cells emerged from a number of studies. It was shown, indeed, that BCR/Abl-positive cells undergo intense glycolysis and glutaminolysis [33], that glycolytic cancer cells in general consume more glutamine than their normal counterparts [34] and that the amount of glucose converted to lactate is increased when murine K-Ras-transformed fibroblasts are grown in the presence of glutamine [35]. Finally, it was shown that OxPhos inhibition promotes glutamine utilization and thereby its biological action(s) [36]. The latter finding is of special relevance to our study, as we incubated cells at very low oxygen tension, to reproduce a critical feature of SCNs in vivo.

The strong relationship between glutamine availability and glucose catabolism in low oxygen emerged from the fact that, while BCR/Abl suppression was inhibited or markedly delayed in the absence of glutamine in parallel with a reduced glucose consumption, the inhibition of this consumption via treatment with 2-DG or 3PO prevented the BCR/Abl-suppressive effects of the presence of glutamine. That the effects of glutamine could be overridden also by PPP inhibition strengthens the conclusion that it is glucose consumption per se, independently of the catabolic pathway driving consumption, that is the pivotal factor controlling BCR/Abl expression/signaling in low oxygen, while glutamine plays the role of the upstream regulator of the phenomenon. That glutamine is coupled to, and acts upstream of, glucose utilization emerged from the demonstration that glutamine up-regulates the MondoA:Mlx complex, a transcriptional repressor of TXNIP, that, in turn, inhibits glucose uptake [37]. While an upstream glutamine-dependent control of glucose uptake explains the effects of the downstream inhibition of glucose catabolism well, the connections between glucose consumption and the further downstream, terminal mechanisms driving BCR/Abl suppression were the object of previous work [38].

The most interesting aspect of BCR/Abl suppression in low oxygen is its relationship to the selection of a cell subset endowed with stem/progenitor cell potential and capable of persisting in culture in the absence of BCR/Abl signaling [11,12,29,30]. The data presented here allowed us to trace this selection back to glutamine availability. On the contrary, the absence of glutamine, apparently via the enhanced persistence of glucose in culture medium, favored the maintenance of BCR/Abl-dependent stem/progenitor cell potential. The prompt availability of BCR/Abl signaling upon transfer to LC2 made it possible to actualize this potential rapidly upon cell transfer to clonal expansion-permissive LC2. While these features appear adequate for the BCR/Abl-expressing/TKi-sensitive LSCs detected in a number of studies, the incapacity, due to BCR/Abl suppression, to exploit the expansion potential rapidly upon transfer to LC2 suits an LSC subset refractory to BCR/Abl inhibitors very well [11,12,29,30]. 

The resistance of both LSC phenotypes to low oxygen in vitro suggests that both are suited to homes within the low-oxygen SCN of BM. The two LSC subsets most likely differ, instead, as for their placement within the different SCN zones we proposed to be characterized, in addition to the lack of oxygen, by the availability or shortage of nutrients such as glucose or glutamine. That the BCR/Abl-positive phenotype adequately defines the LSCs committed to clonal expansion and the BCR/Abl-negative phenotype LSCs capable of long-term persistence as stem cells and, therefore, the maintenance of TKi-resistant MRD, was discussed in detail elsewhere. Previous studies showed that oxygen concentrations as low as 0.2% are compatible with cell survival and proliferation [39] and unpublished data from our laboratory and addressed the question of how LSC maintenance in the absence of BCR/Abl signaling can be sustained in vivo by physiological stimuli released within the SCN [13,27,40,41].

Glutamine emerged from this study as a crucial element for the definition of the metabolic profile of LSCs hosted in the two different SCN zones we referred to above. Cells where BCR/Abl expression is maintained because the glucose concentration is above a certain threshold are to be considered characterized by a “glycolytic” metabolic profile. The data we obtained make it straightforward to hypothesize that the lack of glutamine contributes to sustain glucose availability in the “glycolytic” zone of SCN and facilitates, therein, BCR/Abl-dependent clonal expansion. On the contrary, glutamine-rich SCN zones where glucose is rapidly exhausted would be adequate to select BCR/Abl-negative, non-glycolytic cells, the clonal expansion of which is restrained to the advantage of a prolonged maintenance of stem cell potential [42]. On this basis, we propose glutamine availability as a feature of the SCN zone hosting TKi-resistant MRD of CML.

It is likely that the different SCN zones influence each other, establishing a sort of metabolic symbiosis [13] where a high rate of glucose consumption in the “glycolytic” zones may determine a decrease in pH and an increase in lactate availability in neighboring glucose-free zones. In these zones, lactate and glutamine may ensure an energetic level sufficient for the maintenance of the BCR/Abl-independent cell subset [13], while lactic acidosis may eventually inhibit HIF and induce an “oxidative” metabolic profile, as demonstrated in solid tumors [43,44]. In this respect, it is worth pointing out that (i) glutamine availability has been put in relation to oxidative metabolism, and oxidative glutamine metabolism can even be upregulated in low oxygen despite the decreased mitochondrial respiration [41]; and (ii) the stem cells of a number of cancers, including leukemias, have been shown to rely more on OxPhos and less on glycolysis for energy production [45,46]. In our study, glutamine was found necessary for, or to markedly enhance, the expression of markers of respiration and mitochondrial biogenesis at the end of incubation in low-oxygen LC1, when delayed LC2-repopulating stem/progenitor cell potential is selected. To discuss in detail the question of to what extent an “oxidative” metabolic profile is compatible with cell homing in an environment at very low oxygen tension is well beyond the limits of this study. However, OxPhos inhibition by metformin in low oxygen did exhibit an effect, although on KCL22 cells only (Figure 4F), delaying LC2 repopulation by cells transferred from glutamine-supplemented LC1, i.e., unable to carry on glycolysis because of a glucose shortage. 

BPTES treatment of low-oxygen CML cell cultures yielded results similar to those of the inhibitors of glucose metabolism, indicating that the conversion of glutamine to glutamate controls glucose consumption and the consequent BCR/Abl suppression. Glutamine metabolism in our experimental system is the object of current work, directed first to clarify the roles of TCA cycle anaplerosis with glutamine and reductive glutamine metabolism [16,36], together with their possible connection to the regulation of the glucose uptake mentioned above. Besides the obvious interest of these fundamental studies, we also undertook experiments directed to explore the potential translational value of the results obtained with BPTES. The development of glutaminase inhibitors in view of their therapeutic use has been recently and extensively reviewed [47]. The results obtained indicated that the treatment with IM alone had negligible effects on the level of LC2 repopulation, as expected due to BCR/Abl suppression at the end of LC1 in low oxygen, whereas the co-treatment of IM with BPTES or TGS markedly decreased the maintenance of stem cell potential, pointing to the potential value of this combination for a therapeutic approach to the reduction in or suppression of MRD in CML.

## 5. Conclusions

This study showed that glutamine is a critical regulator of BCR/Abl expression and signaling in low oxygen, via the control of glucose availability. On this basis, we propose the presence of glutamine to characterize bone marrow sites where BCR/Abl-negative, TKi-resistant LSCs of CML are hosted, i.e., where MRD is maintained. The potential translational value of the inhibition of glutamine metabolism to reduce or suppress the maintenance of stem cell potential was tested successfully by co-treating CML cells with glutaminase inhibitors together with TKi.

## Figures and Tables

**Figure 1 cancers-13-04372-f001:**
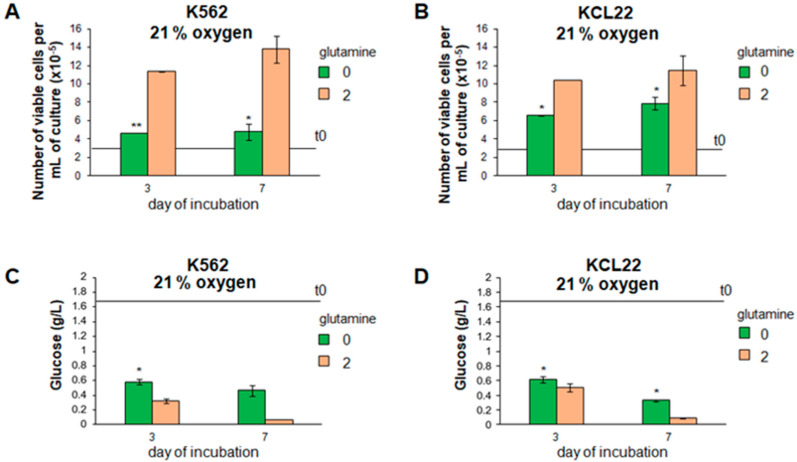
Effects of the lack of glutamine on cell growth and glucose consumption in “normoxia”. K562 (**A**,**C**) or KCL22 (**B**,**D**) cells were seeded at 3 × 10^5^ cells/mL and incubated in standard atmosphere (21% O_2_) in the absence or the presence of glutamine from time 0 (t0) of incubation. Trypan blue-negative cells were counted (**A**,**B**) and glucose concentration in culture medium was measured (**C**,**D**) at the indicated times. Horizontal bars indicate t0 values. Values are mean ± S.D. of data obtained from 3 independent experiments; 0 vs. 2 mM glutamine: * *p* < 0.05, ** *p* < 0.01.

**Figure 2 cancers-13-04372-f002:**
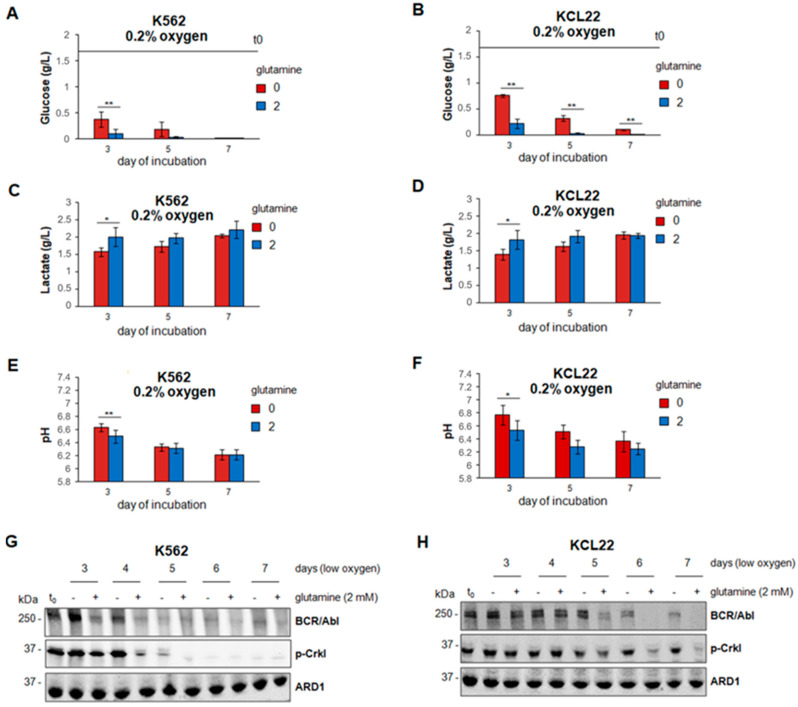
Effects of the lack of glutamine on glucose consumption, lactate production, pH reduction and BCR/Abl expression/signaling in low oxygen. K562 (**A**,**C**,**E**,**G**) or KCL22 (**B**,**D**,**F**,**H**) cells were seeded at 3 × 10^5^ cells/mL and incubated in atmosphere at 0.2% O_2_ for the indicated times in the absence (0/−) or the presence (2/+) of 2 mM glutamine from time 0 (t0) of incubation. (**A**–**F**) Glucose and lactate concentrations and pH were measured in culture medium. Values are mean ± S.D. of data obtained from 3 independent experiments; 0 vs. 2 mM glutamine: * *p* < 0.05, ** *p* < 0.01. Horizontal bars indicate the t0 values. (**G**,**H**) Total cell lysates in Laemmli buffer were obtained at the indicated times of incubation and subjected to SDS-PAGE and immuno-blotting with anti-Abl or anti-p-Crkl Ab; anti-ARD1 Ab was used to verify the equalization of protein loading. One representative experiment out of 3 with similar results is shown. The original Western blotting images and the relative densitometry values can be found in Appendix A.

**Figure 3 cancers-13-04372-f003:**
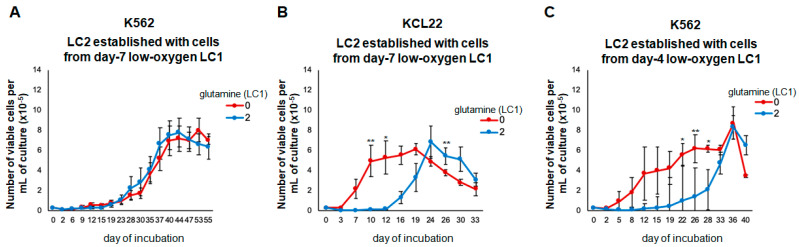
Effects of the lack of glutamine on the maintenance of stem cell potential in low oxygen. K562 (**A**,**C**) or KCL22 (**B**) cells were seeded at 3 × 10^5^ cells/mL and incubated in atmosphere at 0.2% O_2_ in the absence (0) or the presence (2) of 2 mM glutamine from time 0 of incubation (LC1). Cells were rescued from LC1 on day 7 (**A**,**B**) or 4 (**C**) and replated at 3 × 10^4^ cells/mL in growth-permissive, “expansion” secondary cultures (LC2) incubated at 21% O_2_ and in any case in the presence of glutamine. Trypan blue-negative cells were counted at the times of incubation in LC2 indicated in abscissa. Values are mean ± S.D. of data obtained from 3 independent experiments; 0 vs. 2 mM glutamine: * *p* < 0.05, ** *p* < 0.01.

**Figure 4 cancers-13-04372-f004:**
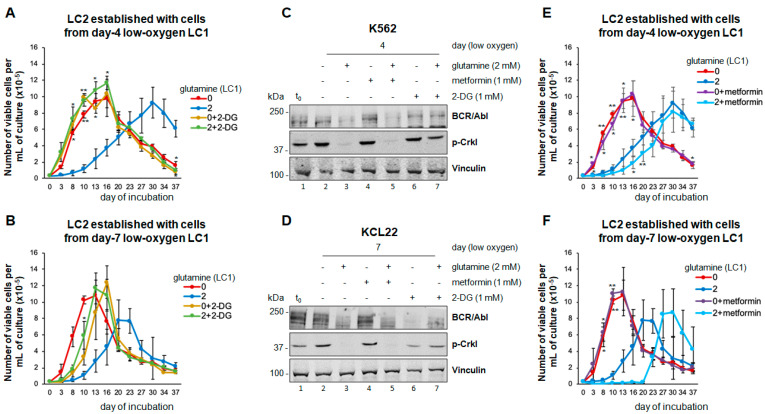
Effects of 2-DG or metformin on BCR/Abl expression/signaling and maintenance of stem cell potential in low oxygen in the absence or the presence of glutamine. K562 (**A**,**C**,**E**) or KCL22 (**B**,**D**,**F**) cells were seeded at 3 × 10^5^ cells/mL and incubated in atmosphere at 0.2% O_2_ for 4 or 7 days, respectively, in the absence (0/−) or the presence (2/+) of 2 mM glutamine and with or without (−/+) the indicated compounds, added at time 0 of incubation (LC1). (**A**,**B**,**E**,**F**) LC1 cells were replated at 3 × 10^4^ cells/mL into drug-free LC2 incubated at 21% O_2_ and in any case in the presence of glutamine. Trypan blue-negative cells were counted at the times of incubation in LC2 indicated in abscissa. Values are mean ± S.D. of data obtained from 3 independent experiments; * *p* < 0.05, ** *p* < 0.01 vs. 2 mM glutamine/no drug. (**C**,**D**) Total cell lysates in Laemmli buffer were subjected to SDS-PAGE and immuno-blotting with anti-Abl or anti-p-Crkl Ab; anti-vinculin Ab was used to verify the equalization of protein loading. One representative experiment out of 3 with similar results is shown. The original Western blotting images and the relative densitometry values can be found in Appendix A.

**Figure 5 cancers-13-04372-f005:**
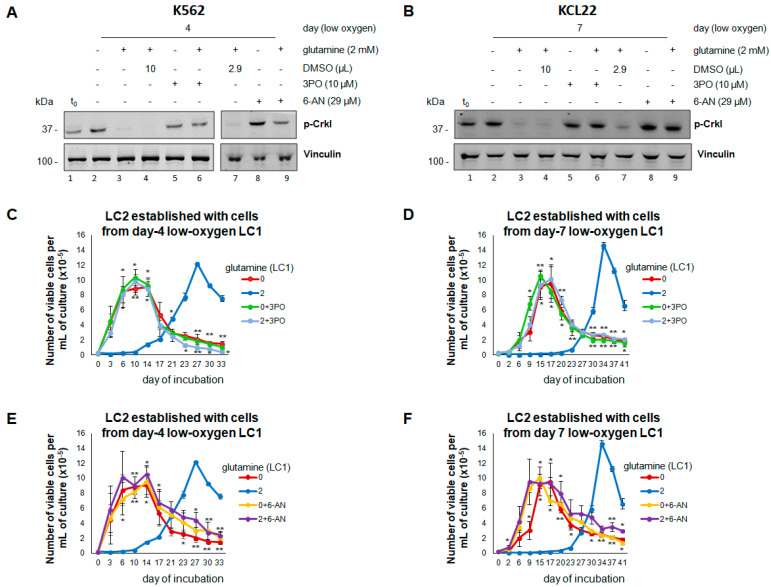
Effects of 3PO or 6-AN on BCR/Abl signaling and maintenance of stem cell potential in low oxygen in the absence or the presence of glutamine. K562 (**A**,**C**,**E**) or KCL22 (**B**,**D**,**F**) cells were seeded at 3 × 10^5^ cells/mL and incubated in atmosphere at 0.2% O_2_ for 4 or 7 days, respectively, in the absence (0) or the presence (2) of 2 mM glutamine and with or without (−/+) the indicated compounds, added at time 0 of incubation (LC1). (**A**,**B**) Total cell lysates in Laemmli buffer were subjected to SDS-PAGE and immuno-blotting with anti-p-Crkl Ab or anti-vinculin Ab, used to verify equalization of protein loading. One representative experiment out of 3 with similar results is shown. (**C**–**F**) LC1 cells were replated at 3 × 10^4^ cells/mL into drug-free LC2 incubated at 21% O_2_ and in any case in the presence of glutamine. Trypan blue-negative cells were counted at the times of incubation in LC2 indicated in abscissa. Values are mean ± S.D. of data obtained from 3 independent experiments; * *p* < 0.05, ** *p* < 0.01 vs. 2 mM glutamine/no drug. The original Western blotting images and the relative densitometry values can be found in Appendix A.

**Figure 6 cancers-13-04372-f006:**
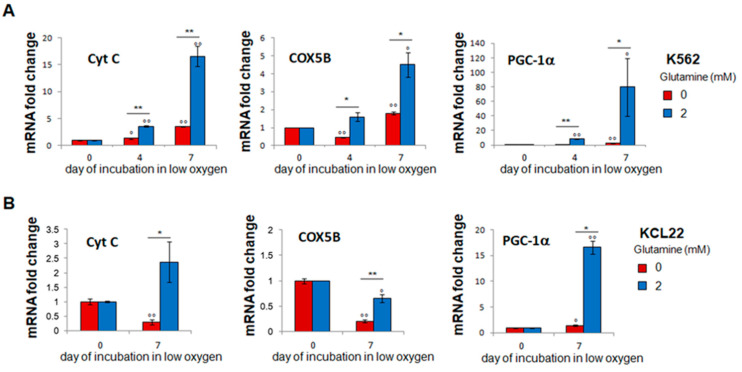
Glutamine enhances the expression of markers of mitochondrial respiration and biogenesis. K562 (**A**) or KCL22 (**B**) cells were seeded at 3 × 10^5^ cells/mL and incubated in atmosphere at 0.2% O_2_ for the indicated times in the absence (0) or the presence (2) of 2 mM glutamine from time 0 of incubation. The expression of cytochrome C (Cyt C), cytochrome-oxidase-5B (COX5B) and PGC-1a mRNA was measured using quantitative real-time PCR. Values are mean ± S.D. of data obtained from 3 independent experiments; ° *p* < 0.05, °° *p* < 0.01 vs. t0; 0 vs. 2 mM glutamine: * *p* < 0.05, ** *p* < 0.01.

**Figure 7 cancers-13-04372-f007:**
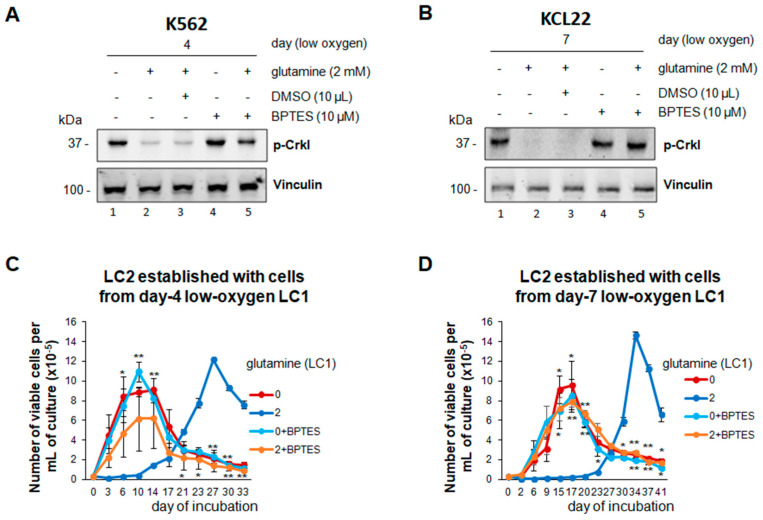
Effects of BPTES on BCR/Abl signaling and maintenance of stem cell potential in low oxygen in the absence or the presence of glutamine. K562 (**A**,**C**) or KCL22 (**B**,**D**) cells were seeded at 3 × 10^5^ cells/mL and incubated in atmosphere at 0.2% O_2_ for 4 or 7 days, respectively, in the absence (0/−) or the presence (2/+) of 2 mM glutamine and with or without (−/+) 10 µM BPTES (**A**–**D**) or its solvent DMSO (A, B) added at time 0 of incubation. (**A**,**B**) Total cell lysates in Laemmli buffer were subjected to SDS-PAGE and immuno-blotting with anti-p-Crkl Ab or anti-vinculin Ab, used to verify equalization of protein loading. One representative experiment out of 3 with similar results is shown. (**C**,**D**) LC1 cells were transferred at 3 × 10^4^ cells/mL into drug-free LC2 incubated at 21% O_2_ and in any case in the presence of glutamine. Trypan blue-negative cells were counted at the times of incubation in LC2 indicated in abscissa. Values are mean ± S.D. of data obtained from 3 independent experiments; * *p* < 0.05, ** *p* < 0.01 vs. 2 mM glutamine/no drug. The original Western blotting images and the relative densitometry values can be found in Appendix A.

**Figure 8 cancers-13-04372-f008:**
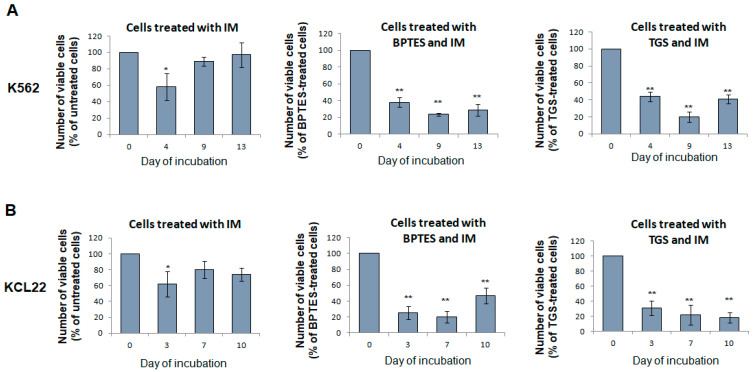
Effect of IM on stem cell potential of CML cells treated or not with glutaminase inhibitors. K562 (**A**) or KCL22 (**B**) cells were seeded at 3 × 10^5^ cells/mL in the presence of glutamine and with or without 10 µM BPTES or 0.5 µM TGS, incubated in atmosphere at 0.2% O_2_ for 4 or 7 days, respectively, and treated with IM 1 µM at day 3 (K562) or 4 (KCL22). LC1 cells were replated at 3 × 10^4^ cells/mL into drug-free LC2 incubated at 21% O_2_ and in any case in the presence of glutamine, and trypan blue-negative cells were counted at the times of incubation in LC2 indicated in abscissa. Plots represent the number of cells treated with IM (left panel), BPTES and IM (central panel), TGS and IM (right panel), expressed as percentages of the number of untreated cells, of cells treated with BPTES alone or of cells treated with TGS alone, respectively. Values are mean ± S.D. of data obtained from 3 independent experiments * *p* < 0.05, ** *p* < 0.01 vs. no IM.

**Table 1 cancers-13-04372-t001:** Primer sequences for PCR.

Gene	Forward	Reverse
b-actin	5′-TCGAGCCATAAAAGGCAACT-3′	5′-CTTCCTCAATCTCGCTCTCG-3′
Cyt-C	5′-TTGCACTTACACCGGTACTTAAGC-3′	5′-ACGTCCCCACTCTCTAAGTCCAA-3′
COX5B	5′-TGCGCTCCATGGCATCT-3′	5′-CCCAGTCGCCTGCTCTTC-3′
PGC-1	5′-GGGAAAGTGAGCGATTAGTTGAG-3′	5′-CATGTAGAATTGGCAGGTGGAA-3′
18S	5′-CGCCGCTAGAGGTGAAATTCT-3′	5′-CGAACCTCCGA CTTTCGTTCT-3′

## Data Availability

All the data relative to this study are presented in the manuscript.

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
