# Peer review of "Glutamine Availability Controls BCR/Abl Protein Expression and Functional Phenotype of Chronic Myeloid Leukemia Cells Endowed with Stem/Progenitor Cell Potential"

_cancers, 2021, doi:10.3390/cancers13174372_

Round 1
Reviewer 1 Report
Minor comments:
1. Simple summary: „a blood cancer” – CML is not a cancer, please use a term “neoplasm”
2. Introduction
“Studies carried out in our laboratory showed that the incubation of CML cells in low oxygen time-dependently suppresses BCR/Abl (but not BCR/Abl, so that BCR/Abl expression can be restored should environmental conditions change) and that a cell subset endowed with stem cell potential is capable to persist in low oxygen independently of BCR/Abl signaling” – please rewrite the sentence; it is not clear for the reader
“This provided a simple explanation for the TKi resistance -the lack of the molecular target of TKi- of cells represent the risk of, and retaining the capacity to sustain, relapse of disease.” - please rewrite the sentence; it is not clear for the reader
3. Reagents: Please check the spelling of 3PO
4. The whole text: please do not use abbreviations, which have not been explained earlier
Reviewer 2 Report
Review
Manuscript number: cancers-1321193
Title: Glutamine availability controls BCR/Abl protein expression and functional phenotype of Chronic Myeloid Leukaemia cells endowed with stem/progenitor cell potential
In this manuscript, the authors could identify the glutamine metabolism as a potential target to overcome the persistence of the therapy-resistant leukemia stem cells (LSC) of chronic myeloid leukemia. The study based on previous results that showed that stem cell niches (SCN) zones with glucose availability are able to predispose LSC to clonal expansion, while zones without glucose availability more likely host the LSC. This means a lack of TKI target but does not affect the gene. Therefore, therapy-resistant MRD can be persist. There is reason to believe that glutamine availability could be influence the BCR/abl expression via the control of glucose consumption from culture medium. The study was able to confirm this.
Major Point:
In my opinion, the study would greatly benefit from the validation of the cell line results (K562 and KCL22) with primary cells from CML patients. In contrast to cell lines, primary cells also reflects all functions and activities in vitro and are therefore much more meaningful relating to the physiological environment. In addition, it is a typical problem that the same experiment with cell lines leads to different phenotype results, depending on the time of cultivation.
Minor Points:
- If possible, please provide a figure legend for the graphical abstract.
- Sentence in the introduction “Studies carried out in our laboratory…. “please specify which BCR/Abl is meaning the protein and which the gene.
- Introduction: There are much more reasons for relapse than the here described features. E.g. occurrence of resistance mutations or other lesions of DNA basis. That should be mentioned.
In summary, a very good and well-written manuscript that should be published in Cancers.
